# Quality of the diagnostic process in patients presenting with symptoms suggestive of bladder or kidney cancer: a systematic review

Yin Zhou,[1] Marije van Melle,[1] Hardeep Singh,[2,3] Willie Hamilton,[4] Georgios Lyratzopoulos,[5] Fiona M Walter[1]

For numbered affiliations see end of article.

**Correspondence to**
Dr Yin Zhou;
ykz21@medschl.cam.ac.uk

## ABSTRACT

**Objectives** In urological cancers, sex disparity exists for survival, with women doing worse than men. Suboptimal evaluation of presenting symptoms may contribute.

**Design** We performed a systematic review examining factors affecting the quality of the diagnostic process of patients presenting with symptoms of bladder or kidney cancer.

**Data sources** We searched Medline, Embase and the Cochrane Library from 1 January 2000 to 13 June 2019.

**Eligible criteria** We focused on one of the six domains of quality of healthcare: timeliness, and examined the quality of the diagnostic process more broadly, by assessing whether guideline-concordant history, examination, tests and referrals were performed. Studies describing the factors that affect the timeliness or quality of the assessment of urinary tract infections, haematuria and lower urinary tract symptoms in the context of bladder or kidney cancer, were included.

**Data extraction and synthesis** Data extraction and quality assessment were independently performed by two authors. Due to the heterogeneity of study design and outcomes, the results could not be pooled. A narrative synthesis was performed.

**Results** 28 studies met review criteria, representing 583 636 people from 9 high-income countries. Studies were based in primary care (n=8), specialty care (n=12), or both (n=8). Up to two-thirds of patients with haematuria received no further evaluation in the 6 months after their initial visit. Urinary tract infections, nephrolithiasis and benign prostatic conditions before cancer diagnosis were associated with diagnostic delay. Women were more likely to experience diagnostic delay than men. Patients who first saw a urologist were less likely to experience delayed evaluation and cancer diagnosis.

**Conclusions** Women, and patients with non-cancerous urological diagnoses just prior to their cancer diagnosis, were more likely to experience lower quality diagnostic processes. Risk prediction tools, and improving guideline ambiguity, may improve outcomes and reduce sex disparity in survival for these cancers.

## Strengths and limitations of this study

► This is the first study to our knowledge that examined factors affecting the diagnostic quality of both kidney and bladder cancer.
► We examined all relevant symptoms in these patients, not limiting to haematuria only.
► We were unable to perform a meta-analysis due to the heterogeneity of the studies.

## INTRODUCTION

Making a correct and timely diagnosis is paramount for patient safety and high quality healthcare. The US National Academy Sciences, Engineering and Medicine (NASEM—formerly the Institute of Medicine) report 'Improving Diagnosis in Health Care' highlights the importance of research on reducing missed and delayed diagnosis and targeting contributory factors that lead to diagnostic errors.[1] Cancer is one of the most common conditions to be affected by diagnostic errors[2] and outpatient malpractice claims.[3] This, in addition to the compelling rationale for early detection, makes cancer an excellent disease model for examining diagnostic safety. Bladder and kidney cancer, two relatively common cancers, pose particular diagnostic challenges. Uniquely among common cancers, women with bladder cancer have poorer survival than men with the same cancer.[4] Missed or delayed referral and diagnosis may contribute to the survival difference between men and women with these urological cancers.[5]

Timeliness, one of the six domains of healthcare quality described by the NASEM, can be regarded as the most relevant for evaluating the diagnostic process in cancer.[1] Timely diagnosis of cancer is important to optimise clinical outcomes and patient experience.[6 7] In the UK, efforts to promote early diagnosis and reducing delays during the diagnostic process have informed many

initiatives aiming to improve outcomes for cancer patients.[8]

We performed a systematic review to examine the factors affecting the quality of the diagnostic process, in particular timeliness, for patients presenting with urological symptoms that may be suggestive of kidney or bladder cancer. Our secondary aim was to examine existing definitions for timeliness of evaluation, referral and diagnosis for these patients.

## METHODS
### Search strategy and study inclusion
We searched Ovid Medline and Embase for relevant from 1 January 2000 to 29 January 2018, with an updated search on 13 June 2019 of both databases and a new search of the Cochrane Library from inception to the same date. We did not restrict on publication type or languages (online supplementary appendix 1). We restricted our search to studies published from 2000 due to prior knowledge that there were few early relevant studies,[9] and that the quality of the diagnostic process for cancer might have been affected by the introduction of national initiatives such as the fast-track referral pathways in the UK in 2000.

We focused on clinical features listed in the English 2015 National Institute for Health and Care Excellence guidelines for suspected cancer[10] in order to examine the population that are most likely to have cancer. We based our outcome measures of diagnostic timeliness on internationally accepted definitions of the diagnostic intervals: for example, primary care interval=time from the patient's first presentation to a primary care practitioner (PCP), to referral.[11 12] We also examined the quality of the diagnostic process more broadly, by assessing whether appropriate or guideline-concordant history, examination, diagnostic tests and referrals were performed during the evaluation of symptoms.

All titles and abstracts were screened by YZ, with 10% of a random selection independently assessed by a second reviewer (MM). Both authors then independently assessed the full-text articles after screening of titles and abstracts. Consensus was sought from GL and FW where disagreements arose.

Inclusion criteria:
► Studies describing the factors that affect the timeliness or quality of the assessment of the following clinical features in the context or bladder or kidney cancer:
Urinary tract infections (UTIs).
Haematuria.
Lower urinary tract symptoms (including dysuria, urinary frequency, urgency, incontinence and nocturia).

Exclusion criteria:
► Studies only describing population or patients under the age of 18 years.
► Conference abstracts, correspondence, editorials, short reports and the grey literature.
► Case reports or case series of <10 patients.

### Data extraction and quality assessment
YZ and MM independently performed data extraction, using a data collection template, on study characteristics, diagnostic intervals, frequency of evaluations, and the patient, clinician and system factors affecting the diagnostic intervals and frequency of evaluations. Quality appraisal was performed using a modified version of the critical appraisal skills programme checklist for cohort studies by both authors (table 1 footnote).[13] Any disagreements were resolved by discussion with all members of the research team.

### Data synthesis and analysis
We were unable to pool the results due to the heterogeneity of the study design and outcomes. A narrative synthesis was therefore performed.

### Patient and public involvement
No patient was involved in this review.

## RESULTS
### Study characteristics and quality
Twenty-eight papers, representing 583 636 people, were included after full-text reviews (figure 1). All studies were from high-income countries. These include 18 from the USA, two from Australia, two from the UK, two from Sweden and one each from Finland, Canada, Austria and Italy, and Germany and Austria (in one study). Six examined cancer patients with no predefined clinical features (five bladder, one both bladder and kidney), five focused on patients with haematuria (one of which included only visible haematuria (VH)), eight examined bladder cancer patients with haematuria and one focused on upper urothelial tract cancer patients with haematuria. Eight studies were carried out in the primary care setting, 12 in hospital and 8 in both (table 1).

The main bias and applicability concerns related to the suboptimal identification and/or adjustment for confounders in 18 of the studies, 6 of which were studies using questionnaires,[14–19] 10 were retrospective cohort studies providing descriptive statistics mainly, using record reviews (n=5)[20–24] and electronic health records (n=5),[25–29] 1 was a case–control study[30] and 1 an ecological study.[31]

### Quality of diagnostic process
#### Diagnostic timeliness
Seventeen of the 28 included studies described diagnostic intervals for patients with either urological symptoms or who had been diagnosed with bladder or kidney cancer (table 2).

Definitions of timely evaluation, referral and diagnosis, were described in 12 studies. For time to first evaluation including cystoscopy, upper urinary tract imaging or urine cytology, Garg et al used a threshold of 30 days,[32] while two studies examined proportions of patients undergoing these tests within 60[33] and 90 days.[24] The remaining studies

**Table 1** Study characteristics of included publications and quality assessment

| Papers | Year/s of data collection | Design | Setting (primary care, specialist or both) | Sample size | Population | CASP quality assessment items | | | | | | | | |
|---|---|---|---|---|---|---|---|---|---|---|---|---|---|---|
| | | | | | | A | B | C | D | E | F | G | H | I |
| Ark et al[34] 2017, USA | 2001–2009 | Retrospective cohort study | Primary care | 1412 | Patients aged 40+ years with bladder cancer and haematuria | Yes | Yes | Yes | Yes | Yes | Yes | Yes | Yes | Yes |
| Aziz et al[14] 2015, Germany and Austria | 2010–2013 | Retrospective questionnaire | Specialist | 68 | Patients undergoing TUR-BT for newly diagnosed urothelial carcinoma of bladder | Unclear | No | Unclear | No | No | No | No | No | Unclear |
| Bassett et al[35] 2015, USA | 2009–2010 | Retrospective cohort study | Primary care | 9211 | Patients with non-visible haematuria seen by a PCP in an outpatient setting, age 65+ years | Yes | Yes | Yes | Yes | Yes | Yes | Yes | Yes | Yes |
| Blick et al[20] 2010, UK | 1997–2006 | Retrospective record review | Specialist | 200 | Consecutive patients with newly diagnosed bladder cancer identified from the records of MDT meetings and theatre records | Yes | No | No | Yes | No | No | No | No | Yes |
| Bradley et al[21] 2016, USA | 2012–2014 | Retrospective record review/ cross-sectional cohort | Specialist | 237 | Women >55 years with asymptomatic microhaematuria | Yes | Yes | Yes | Unclear | Unclear | No | Unclear | No | Yes |
| Buteau et al[25] 2014, USA | 2009–2010 | Retrospective record review | Primary care | 449 | Patients with over 5 RBC/HPF, with both gross and microscopic haematuria | Yes | Yes | Yes | Yes | Yes | No | Unclear | No | Yes |
| Chappidi et al[37] 2017, USA | 2010–2014 | Retrospective claims review | Both | 1326 | Patients with UTUC with a haematuria claim 1 year before diagnosis, under 65 years, 4.4% (n=58) with concomitant bladder cancer diagnosis at time of UTUC diagnosis | Yes | Yes | Yes | Yes | Yes | Yes | Yes | Yes | Yes |
| Cohn et al[39] 2014, USA | 2004–2010 | Retrospective claims review | Both | 7649 | Patients who had an initial haematuria claim within 12 months of an initial bladder cancer claim, age <66 years (upper limit of MarketScan Database population) | Yes | Yes | Yes | Yes | Yes | Yes | Yes | Yes | Yes |
| Elias et al[38] 2010, USA | 2006–2007 | Retrospective record review | Primary care | 164 | Participants with microscopic haematuria on urine dipstick testing or 3+ RBC/HPF on urinalysis. Recruited from well patient clinics aged 50+ years with a 10-year or greater smoking history (any number of cigarettes) and/or a significant (15 or more years) high-risk occupation (such as working in the dye, petroleum or chemical industry) | Yes | Yes | | | Yes | Yes | Yes | Yes | Yes |
| Friedlander et al[36] 2014, USA | 2004–2012 | Retrospective cohort review | Primary care | 2455 | Patients aged 40+ years with first episode of haematuria between 2004 and 2012 either by urinalysis (>3 RBC/HPF) or ICD diagnosis codes for haematuria | Yes | Yes | Unclear | Yes | Yes | Yes | Unclear | Unclear | Yes |
| Garg et al[32] 2014, USA | 2000–2007 | Retrospective cohort study | Both | 35 646 | Patients aged 66+ years with primary bladder cancer with a haematuria claim in 12 months before diagnosis | Yes | Yes | Yes | Yes | Yes | Yes | Yes | Yes | Yes |

Continued

**Table 1** Continued

| Papers | Year/s of data collection | Design | Setting (primary care, specialist or both) | Sample size | Population | CASP quality assessment items | | | | | | | | |
|---|---|---|---|---|---|---|---|---|---|---|---|---|---|---|
| | | | | | | A | B | C | D | E | F | G | H | I |
| Han et al[31] 2018, USA | 2014 | Cross-sectional, ecological study | Specialist | 306 Hospital Referral Regions | Medicare beneficiaries aged 65–99 years with bladder cancer and underwent at least one cystoscopy procedure; consisting of 173551 female and 286090 men in total | 😊 | 😊 | ☹️ | 😐 | ☹️ | ☹️ | 😐 | 😐 | 😊 |
| Henning et al[15] 2013, Austria and Italy | Not mentioned | Prospective questionnaire study | Specialist | 200 | Consecutive series of 200 patients admitted for elective TUR-BT | 😊 | 😊 | ☹️ | 😐 | ☹️ | ☹️ | 😐 | 😐 | 😊 |
| Hollenbeck et al[26] 2010, USA | 1992–2002 | Retrospective cohort study | Both | 29 740 | Patients with haematuria 1 year prior to bladder cancer diagnosis; 66+ years | 😊 | 😊 | 😊 | 😊 | 😊 | ☹️ | 😐 | 😐 | 😊 |
| Johnson et al[27] 2008, USA | 1998–2002 | Retrospective cohort study | Both | 926 | Patients with newly diagnosed haematuria, 18+ years | 😊 | 😐 | 😊 | 😐 | 😐 | 😐 | 😐 | 😊 | 😊 |
| Liedberg et al[16] 2016, Sweden | 2014–2015 | Interventional study | Both | 376 | 275 patients in intervention group, 101 in control group; aged 50+ years with macroscopic haematuria | 😊 | 😐 | 😊 | 😊 | ☹️ | ☹️ | 😐 | 😊 | 😊 |
| Lyratzopoulos et al[40] 2013, UK | 2009–2010 | Secondary analysis of audit data | Primary care | 1318 | Bladder and kidney cancer patients | 😊 | 😊 | 😊 | 😊 | 😊 | 😊 | 😊 | 😊 | 😊 |
| Matulewicz et al[28] 2019, USA | 2007–2015 | Retrospective cohort study | Multi-institutional hospital system | 15 161 | Microscopic haematuria in patients aged 35 years and over | 😊 | 😊 | 😊 | 😊 | 😊 | 😐 | 😊 | 😊 | 😊 |
| McCombie et al[17] 2017, Australia | 2008–2014 | Retrospective cohort study, telephone survey | Specialist | 100 | Bladder cancer patients | 😊 | 😊 | 😐 | 😊 | ☹️ | ☹️ | 😐 | ☹️ | 😊 |
| Murphy et al[23] 2017, USA | 2012–2014 | Retrospective cohort study | Primary care | 495 | Patients with haematuria (urine red blood cells of >50 cells per high power field) | 😊 | 😊 | 😊 | 😊 | ☹️ | ☹️ | 😐 | 😐 | 😊 |
| Ngo et al[41] 2017, Australia | 2015 (12-month period) | Retrospective record review | Specialist | 305 | Cystoscopy cases primarily for haematuria investigation, 18+ years | 😊 | 😊 | 😊 | 😊 | 😊 | 😊 | 😊 | 😊 | 😊 |
| Nieder et al[18] 2010, USA | Not mentioned | Questionnaire study | Primary care | 788 | 788 PCPs (internal medicine, family practice, primary care or obstetrics and gynaecology) randomly selected from the Little Blue Book of Miami-Dade County and Dallas, published by National Physicians Data Source | 😊 | 😊 | 😐 | ☹️ | ☹️ | ☹️ | 😐 | 😐 | |
| Richards et al[24] 2016, USA | 2007–2009 | Retrospective cohort study | Both | 12 195 | Patients from SEER-Medicare with a haematuria or UTI claim; 66+ years | 😊 | 😊 | 😊 | 😊 | 😊 | 😊 | 😊 | 😊 | 😊 |
| Nilbert et al[30] 2018, Sweden | 2015–2016 | Case-control study | Secondary care | 1697 controls/174 cases | Patients with macroscopic haematuria (control) and bladder and upper urothelial tract cancer (cases), aged 40 years and over | 😊 | 😊 | 😐 | 😊 | 😐 | ☹️ | 😊 | 😊 | 😊 |
| Richards et al[24] 2018, USA | 2011–2013 | Retrospective record review | Both | 201 | Consecutive patients with new-onset haematuria, 195 male | 😊 | 😐 | 😊 | 😊 | 😐 | 😊 | 😊 | 😐 | 😊 |

Continued

**Table 1** Continued

| Papers | Year/s of data collection | Design | Setting (primary care, specialist or both) | Sample size | Population | A | B | C | D | E | F | G | H | I |
|---|---|---|---|---|---|---|---|---|---|---|---|---|---|---|
| | | | | | | CASP quality assessment items | | | | | | | | |
| Santos et al[29] 2015, Canada | 2000–2009 | Retrospective cohort review | Specialist | 1271 | Patients who underwent radical cystectomy for bladder cancer, had a first urologist visit after having visited a GP or ED physician, aged >40+ years | green | amber | amber | amber | amber | red | amber | red | green |
| Sell et al[19] 2019, Finland | Unknown | Substudy of RCT[ix], using questionnaires | Specialist | 131 | Patients with histologically proven low-grade bladder cancer with self-reported macroscopic haematuria | green | green | green | amber | red | red | amber | green | green |
| Shinagare et al[22] 2014, USA | 2004–2012 | Retrospective record review | Specialist | 100 | Consecutive patients with asymptomatic haematuria | green | amber | amber | green | red | amber | amber | red | amber |

CASP Quality Assessment Items (Key: green=yes, yellow=can't tell, red=no).
A: Does the study address a clearly focused issue?
B: Was the cohort recruited in an acceptable way?
C: Was the exposure accurately measured to minimise bias?
D: Was the outcome accurately measured to minimise bias?
E: Have the authors identified all important confounding factors?
F: Have they taken account of the confounding factors in the design and/or analysis?
G: Do you believe the results?
H: Can the results be applied to the local population?
I: Do the results of this study fit with the other available evidence?
CASP, critical appraisal skills programme; ED, emergency department; GP, general practitioner; MDT, multidisciplinary team; PCP, primary care practitioner; RBC/HPF, red blood cell per high power field; RCT, randomised controlled trial; TUR-BT, transurethral resection of bladder tumour; UTI, urinary tract infection; UTUC, upper tract urothelial carcinoma.

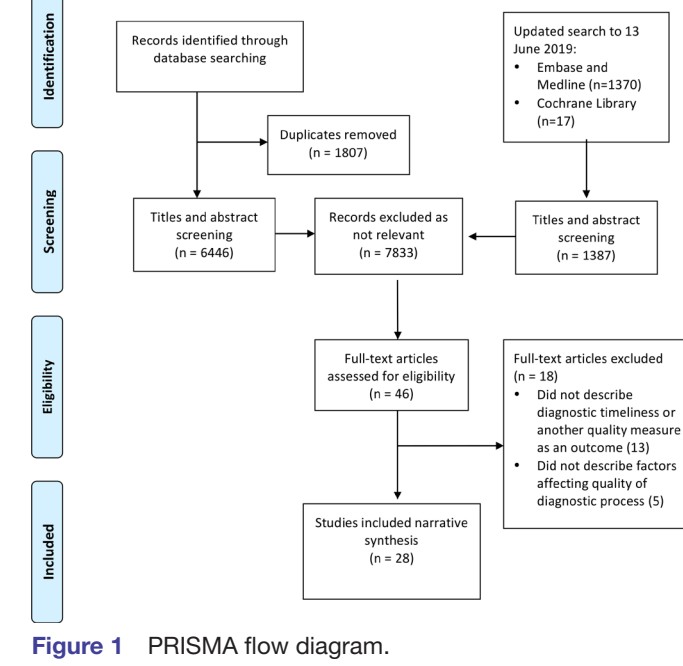

**Figure 1** PRISMA flow diagram.

used 180 days as time cut-offs for which they considered evaluation should be carried out,[22 28 34–36] although one also looked at completion within 365 years, and beyond.[28] Thresholds for referral was set at 90 days by one paper,[27] while delays in cancer diagnosis were defined as greater than 90 days[37] and in 3 month increments up to 1 year.[26 28]

## Other quality dimensions

We found no standard definition for high quality care during the diagnostic process in any of the included studies. Most studies reported the frequency of appropriate or guideline-concordant diagnostic tests and referrals performed during diagnostic evaluation (online supplementary appendix 2). Studies examining the frequency of non-evaluation of haematuria reported this to be 47%–81% within 60 days of initial symptom presentation,[24 33] reducing to 36%–65% in studies by 180 days.[22 34–36]

Eleven studies reported the percentages of investigations and referrals performed in patients with haematuria,[18 21 22 24 25 28 33–36 38] seven of which also specified time-frames during which these evaluations should be completed[22 24 28 33–36] (online supplementary appendix 2). Five studies reported that only 5%–25% of these patients received both imaging and cystoscopy (commonly defined as 'complete evaluation' by the studies) by 6 months of their first presentation with haematuria[25 28 34–36] and case series of 100 patients in a single institution reported this percentage to be 64% in their cohort.[22]

Studies reported variations in the percentages of patients with haematuria who received urine culture (15%–84%), urine cytology (5%–43%), imaging tests (14%–76%) and cystoscopy (6%–26%) at at least 2 months after presentation, indicating that there were variations in how clinicians evaluate patients with haematuria. In studies that focused

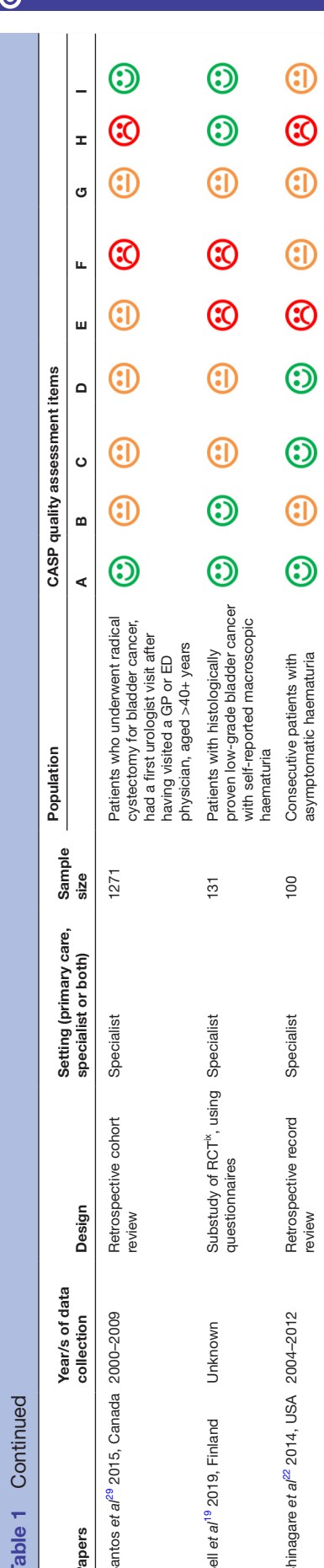

**Table 2** Mean and median diagnostic intervals from first presentation to diagnosis as described in included studies

| Study | Population of interest† | Time interval‡ | Event points (shading denotes interval) | Mean interval duration (days) All | Mean Men | Mean Women | Median interval duration (days) All | Median Men | Median Women |
|---|---|---|---|---|---|---|---|---|---|
| Aziz et al[14] 2015, Austria/Germany | VH§ | T1 | | 335 | 343.70 | 313.29 | | | |
| Chappidi et al[37] 2017, USA | UTUC§ with haematuria | T1 | | | 93.5* | 84.4* | | 60* | 49* |
| Cohn et al[39] 2014, USA | Bladder cancer with haematuria | T1 | | | 85.4* | 73.6* | | 41* | 35* |
| Liedberg et al[16] 2016, Sweden | VH§ | T1 | | | | | 29 versus 50* (intervention vs control) | | |
| Nilbert et al[30] 2018, Sweden | Bladder/ UTUC with VH | T1 | | | | | 25 versus 35* (intervention vs control) | | |
| Richards et al[42] 2016, USA | Bladder | T1 | | | 58.9* | 72.2* | | | |
| Garg et al[32] 2014, USA | Bladder cancer with haematuria | T2 | | | | | 8 | 8 | 9 |
| Matulewicz et al[28] 2019, USA | VH | T2 | Time to imaging | | | | 75 | | |
| Matulewicz et al[28] 2019, USA | VH | T2 | Time to cystoscopy | | | | 68.5 | | |
| Garg et al[32] 2014, USA | Bladder cancer with haematuria | T3 | | 27 | 24* | 35* | 3 | 2* | 6* |
| Richards et al[24] 2018, USA | NVH§ | T3 | | | | | 28 | | |
| Santos et al[29] 2015, USA | Bladder | T3 | | | | | 30 | 23* | 56* |
| Chappidi et al[37] 2017, USA | UTUC§ with haematuria | T3 | | | 17.9 | 20.4 | | 4 | 5 |
| Johnson et al[27] 2008, USA | Haematuria | T4 | | | | | 33.5 | 27.4* | 36.5* |
| Liedberg et al[16] 2016, Sweden | VH§ | T4 | | | | | 14 versus 33 | (intervention vs control) | |
| Lyratzopoulos et al[40] 2013, UK | Bladder | T4 | | | | | | 4* | 6* |
| Lyratzopoulos et al[40] 2013, UK | Kidney | T4 | | | | | | 10* | 16* |
| McCombie et al[17] 2017, Bladder Australia | Bladder | T4 | | | | | 3 | | |
| Nilbert et al[30] 2018, Sweden | Bladder/ UTUC with VH | T4 | | | | | 0 vs 7 | (intervention vs control) | |

Event points: First presentation with symptom, Referral to specialist, First specialist appointment, First investigation, Diagnosis

Continued

**Table 2** Continued

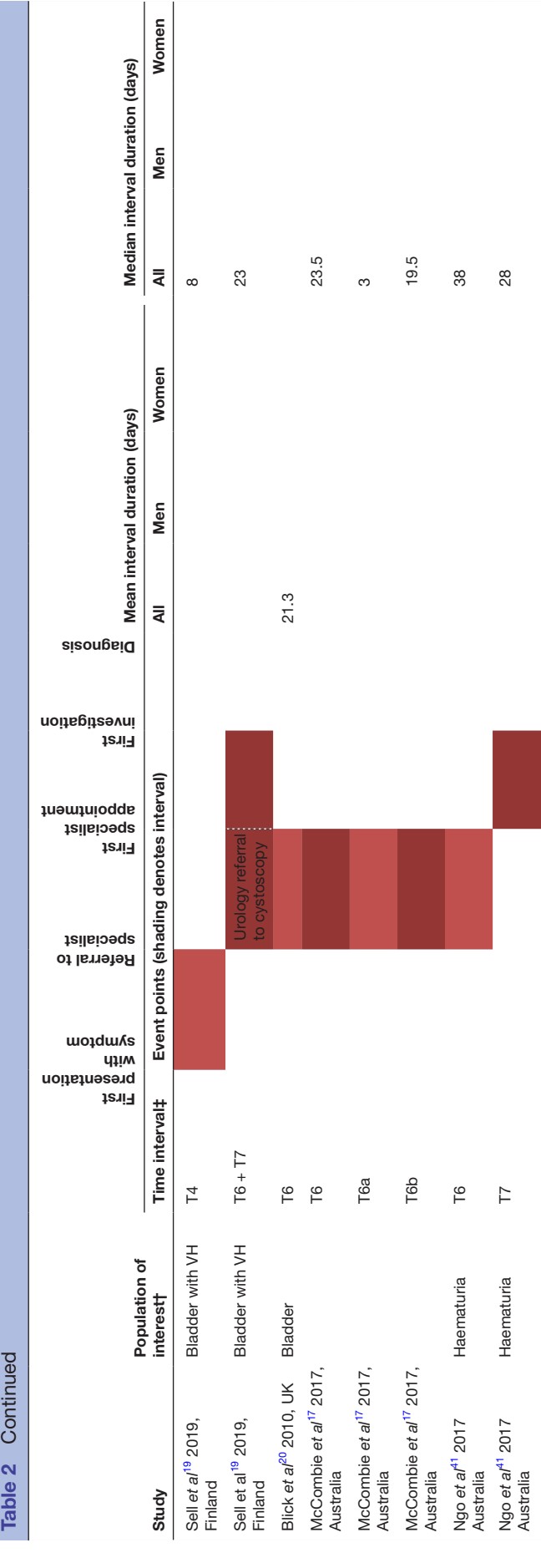

| Study | Population of interest† | Event points (shading denotes interval) | Time interval‡ | Mean interval duration (days) | | | Median interval duration (days) | | |
|---|---|---|---|---|---|---|---|---|---|
| | | | | All | Men | Women | All | Men | Women |
| Sell et al[19] 2019, Finland | Bladder with VH | | T4 | | | | 8 | | |
| Sell et al[19] 2019, Finland | Bladder with VH | Urology referral to cystoscopy | T6 + T7 | | | | 23 | | |
| Blick et al[20] 2010, UK | Bladder | | T6 | 21.3 | | | | | |
| McCombie et al[17] 2017, Australia | | | T6 | | | | 23.5 | | |
| McCombie et al[17] 2017, Australia | | | T6a | | | | 3 | | |
| McCombie et al[17] 2017, Australia | | | T6b | | | | 19.5 | | |
| Ngo et al[41] 2017, Australia | Haematuria | | T6 | | | | 38 | | |
| Ngo et al[41] 2017, Australia | Haematuria | | T7 | | | | 28 | | |

*P value ≤0.05 for difference between men and women.

†Population of interest indicates the characteristics of the cohort examined in each study (either symptom or type of cancer patients).

‡Key for time intervals: T1, time from first presentation with symptom to diagnosis; T2, time from first presentation with symptom to first investigation; T3, time from first investigation; T4, time from first presentation to referral to specialist; T5, time from referral to specialist to diagnosis; T6, time from referral to specialist to first specialist appointment; T7, time from first specialist appointment to first investigation.

NVH, non-visible haematuria; UTUC, upper tract urothelial carcinoma; VH, visible haematuria.

on the type of haematuria, a larger proportion of patients with VH (25%)[25] received both imaging and cystoscopy than patients with non-visible haematuria (NVH) (up to 14%).[25 28 35] Between 21% and 36% of patients with haematuria received a urological referral,[34 35 38] although a survey study from almost 800 PCPs in the USA reported that about one-third and two-thirds of them would refer patients with NVH and VH, respectively.[18]

One secondary care study reported non guideline-compliant practice in over a third of postmenopausal women with asymptomatic haematuria, with no documentation of full genitourinary examination with vaginal tissue quality and presence of prolapse.[21]

## Presenting symptoms

Ten of the 20 studies which extracted symptom information used coded information from routine or claims data,[21 25 28 29 31 34 35 37–39] 5 used self-reported symptoms from questionnaire and audit data,[14–16 19 40] 6 studies performed direct record review[22 24 25 30 33 41] and 1 used coded information and record review.[17]

Eleven studies examined the direct association between the type of presenting symptoms and quality of the diagnostic process, the majority focusing on haematuria[18 19 22 24 27 28 30 39–41] and one on UTI.[42] No other presenting symptoms were examined. A large population-based study in the UK reported that bladder or kidney cancer patients presenting with haematuria were significantly less likely to have three or more consultations before a general practitioner (GP) referral compared with those who did not present with haematuria (OR 0.29 CI 0.19 to 0.46, p<0.001 for bladder cancer; OR 0.64 CI 0.30 to 1.37, p=0.25 for kidney cancer).[40] An earlier study of about 1000 patients found that there was also a dose-dependent relationship between the number of haematuria visits and the likelihood of a urological referral (HR 5.18 and 7.66 for 2 and 3 visits, respectively, vs 1 visit, p<0.0001),[27] and a high-quality US study involving claims review found that increasing number of haematuria visits was associated with diagnostic delay for bladder cancer patients.[39]

VH predicted a shorter time to evaluation,[39] to referral,[19] a lower likelihood of incomplete evaluation,[22 24] shorter time from GP referral to urology consultation,[41] and a shorter time to diagnosis[39] than NVH.

## Other recent diagnoses preceding cancer diagnosis

Between 20% and 61% of women and 15% and 35% of symptomatic men were treated or diagnosed with a UTI before being diagnosed with bladder cancer.[14 15 37 39] Women are also four times as likely as men to receive three or more courses of treatments for UTIs before their cancer diagnosis (15.8% vs 3.8% for women vs men, p=0.04).[15]

In two case series, it was also reported that a significant proportion of bladder cancer patients (up to 40%) received symptomatic treatments for either lower urinary tract symptoms or abdominal pain before referral to a urologist[14] or were not further evaluated, with women more likely to be affected than men (41.7% vs 16.2% once or twice, 5.6% vs 2.9% three or more times, women vs men; p=0.04).[15]

Two large US studies reported that benign diagnosis (up to 12 months prior to cancer diagnosis) such as UTIs, nephrolithiasis and prostate-related diagnosis were associated with delays in cancer diagnosis.[37 39] UTIs were associated with a twofold increase in the odds of diagnostic delay by at least 3 months in both sexes for both upper tract urothelial cancer and bladder cancer,[37 39] compared with those with no UTI diagnosis prediagnosis (OR 1.97, CI 1.74 to 2.22).[39] This was regardless of whether patients first presented to a urologist or other specialty doctors.[37] Nephrolithiasis (RR 1.29, CI 1.07 to 1.54; p=0.007 vs RR 1.09, 0.81 to 1.47 for men vs women) and benign prostatic conditions (such as prostatitis, benign prostatic hyperplasia and benign prostatic nodule) were more likely to predict diagnostic delay in men than women with upper tract urothelial cancer.[37] In this nationwide insurance claims study, no validation of coded information was performed using medical record review.

## Patient factors

### Sex

The effect of sex on diagnostic activity and timeliness were reported by 15 studies.[14 15 19 22 25 27–29 31 32 34 36 38–40] Most evidence indicated that women were less likely than men to undergo any investigation,[35] have complete evaluation,[28 35] be referred to a specialist[27 29 41] and to have cystoscopy or imaging.[31 36 41] Female sex was also a consistent predictor for delayed evaluation of haematuria[32 34 39] and UTIs,[15 25] and longer diagnostic intervals for cancer.[29 32 40] This sex disparity with respect to evaluation and referral was insignificant for patients with NVH.[22 25 38] One Finnish study using patient questionnaires reported no difference in patient (time from symptom recognition to presentation) and primary care (time from presentation to referral to a urologist) intervals in 131 low-grade bladder cancer patients between men and women.[19]

### Other patient factors

In general, evidence for other patient factors affecting the quality of the diagnostic process was less consistent. Two large US studies consisting of over 65 000 patients in total found that older bladder cancer patients with haematuria had longer delays to evaluation than younger patients.[26 32] Increasing comorbidity predicted slower time to urologist, longer delay to evaluation,[32] and diagnosis[39] in two large US samples. While five of the six studies reported no association between ethnicity and quality of the diagnostic process, one study with about 1400 participants reported that African-American bladder cancer patients were less likely than their Caucasian counterparts to: be referred to a urologist (adjusted OR 0.72; 95% CI 0.56 to 0.93; have a cystoscopy (adjusted OR 0.67; 95% CI 0.50 to 0.89), or have imaging (adjusted OR 0.75; 95% CI 0.59 to 0.95).[34]

**Table 3** Summary of association between patient factors and diagnostic safety and timeliness

| Patient factor | No of studies exploring risk factor | Association between patient factor and diagnostic safety and timeliness | | |
| --- | --- | --- | --- | --- |
| | | Delayed / incomplete evaluation | Delayed referral | Longer diagnostic interval |
| Sex | 15 | Women>men | Women>men | Women>men |
| Increasing age | 12 | NS[24 25 28 38] Positive association[26 32] | NS[19 25 32 41] | NS[39] |
| Ethnicity | 7 | NS[25 26 28 32 35 36] African-American worse[34] | NS[25 26 32 35 36] African-American worse[34] | NS[25 26 32 35 36] |
| SES | 5 | NS[16 26 36 39 41] | | |
| Comorbidity | 4 | Positive association[32] NS[24 37] | | Positive association[39] NS[24 37] |
| Smoking | 6 | NS[25 35 36 41] Positive association[38] | NS[19 25 35 36 41] | |
| Anticoagulant use | 5 | NS[24 35 41] More likely to receive imaging[36] | NS[24 30 35 41] | NS[30] |

NS, statistically non-significant; SES, socioeconomic status.

The evidence between socioeconomic status, comorbidity, smoking and anticoagulant use was either non-significant or weak (table 3).

### Clinician factors
#### Physician type
Six studies from the USA examined the type of clinicians as a predictor for diagnostic delay (table 4). Patients who first saw a urologist for their symptoms were less likely to have a delay in evaluation[32] or cancer diagnosis[37] and more likely to have guideline-adherent evaluation[22] than those who first saw another specialty doctor.

When comparing specialties excluding urology, a mixed pattern was seen with respect to referral and use of investigations. In general, there was little evidence to suggest that family physicians in the USA differ from other specialists with respect to evaluating haematuria. Family physicians may be less likely to refer for VH,[25 27] although evidence for delayed referral in patients with UTI and NVH was less clear.[18 25]

### System factors
#### Diagnostic pathways
Three studies examined the impact of interventions in the diagnostic pathways on diagnostic intervals in the UK and Sweden.[16 20 30] A single institution UK study found that the time from GP referral to first hospital visit shortened from 42.9 to 21.3 days (p<0.001) after the introduction of the fast-track pathway, in which patients with alarm symptoms are typically seen or investigated by a specialist within 2 weeks of a GP referral.[20] In Sweden, the introduction of a telephone hotline for patients with VH reduced the time from haematuria to urology referral (33 to 14 days, p=0.32), referral to diagnosis (19 to 8 days, p=0.003) and total healthcare interval (50 to 29 days, p=0.03).[16] Patients with eligible symptoms were able to access a nurse consultant directly by telephone, who

then scheduled the patient for serum creatinine, urine cytology and appointment with a urologist for flexible cystoscopy and CT urography within 2 weeks, all with the same priority as other patients referred by their GP but bypassing the routine referral system.[16] Another Swedish study studying a similar streamlined diagnostic pathway found that it shortened the diagnostic interval from 35 to 25 days (p=0.01) although time to treatment did not change from a regular referral pathway.[30]

### Other factors
Other factors that were found to impact on the quality of the diagnostic process were described by studies using direct record review. These include patient factors such as not attending, cancelling or declining to attend follow-up appointments,[33] delays in PCPs reviewing results, lack of receipt of referral,[17] and scheduling and coordination delay of follow-up test or appointment,[24 33] although the detailed effects of these factors on the quality of the diagnostic process were not reported.

## DISCUSSION
Our review identified several potential areas of missed opportunities in urological cancer diagnosis; it also provides evidence for informing the development of future interventions and research.

### Non-evaluation of haematuria
Studies reported high frequencies of non-evaluation of haematuria, with about two-thirds of patients having no evaluation up to 180 days after initial presentation. Although we found no consistent definition of diagnostic timeliness for evaluation, referral and diagnosis of urological cancer, high percentages of non-evaluated cases likely harbour missed opportunities for a timely diagnosis. For instance, patients with VH should receive renal function

**Table 4** Associations between physician specialty and quality of diagnostic process for patients presenting with different clinical features precancer diagnosis

| Clinical feature | Delay in evaluation | Delay in referral | Delay in diagnosis |
|---|---|---|---|
| Urinary tract infection (UTI) | No difference between PCP* and other specialists or ED physicians (Buteau) | No difference between PCP* and other specialists or ED physicians (Buteau) | Both urologist and non-urologist (urologist: RR1.74, CI 1.31 to 2.31, p<0.001; non-urologist RR 1.44, CI 1.22 to 1.71, p<0.001) (Chappidi) |
| Microscopic haematuria | No difference between PCP* and other specialists or ED physicians (Buteau); OBGYN less likely to perform imaging than medical counterparts (p<0.004) (Neider); guideline concordant with urologist vs non-urologist (OR 54.7, CI 10 to 102, p<0.0001) (Shinagare) | No difference between PCP* and other specialists or ED physicians (Buteau, Neider) | |
| Macroscopic haematuria | OBGYN less likely to perform imaging than medical counterparts (p<0.01) (Neider) | PCP* less than other specialists or ED physicians (Buteau); no difference between specialties (Neider) | |
| Haematuria (not specified) | Initial visit with urologist associated with reduced odds of delayed evaluation (OR 0.34, CI 0.31 to 0.68, p<0.001) compared with primary care and OBGYN (Garg) | Internal medicine providers and other specialists more likely than family physicians to refer (HR 1.30, 1.03 to 1.64; HR 1.72, 1.01 to 2.90). No difference in hospital specialists from family medicine (Johnson). | |
| Nephrolithiasis | | | Delay in non-urologist versus urologist (RR 1.25, CI 1.05 to 1.49, p=0.01) (Chappidi) |
| Benign prostate conditions | | | Delay in non-urologist versus urologist (new prostate conditions—RR 1.41, CI 1.12 to 1.78, p=0.003); recurrent prostate conditions RR 1.94, CI 1.45 to 2.58, p<0.001) (Chappidi) |

*PCP includes family medicine and internal medicine.
ED, emergency department; OBGYN, obstetricians and gynaecologists; PCP, primary care physician.

testing, imaging and urology referral for cystoscopy once a transient cause such as UTI has been excluded.[43] Patients with persistent NVH should additionally receive a blood pressure check and urinary albumin-creatinine ratio as part of the evaluation.[43] Given that the PPV of haematuria for urological cancer can be as high as 11%,[44] lack of evaluation could lead to missed diagnoses. Improving clinicians' awareness and adherence to existing guidelines as well as using electronic algorithms to flag up abnormal findings[33] may reduce such missed opportunities.[23]

### Women experience poorer quality of diagnostic process than men

Our review found that women with haematuria were more likely to be treated for UTIs or for pain, and less likely to be evaluated further or referred than men. Benign conditions such as UTIs and atrophic vaginitis are commoner in women. At the same time, women are less likely to have bladder and kidney cancer than men (M:F ratio about 3:1 and 1.7:1 for bladder and kidney, respectively).[45] This may be due to differing exposure to lifestyle and environmental factors, and biological propensity to these cancers by sex.[4] The combined effect of greater frequency of non-neoplastic disease in women, and the

fact that urological cancer is less common in women, means that the PPV of relevant symptoms for bladder and kidney cancer is lower in women than men.

Although a relevant urological symptom is more likely to be due to a benign cause in women than in men, avoidable diagnostic delay for urological cancer may still occur if there is a failure to ensure the resolution of symptoms. Current American Urological Association guidelines on the evaluation of asymptomatic microscopic haematuria recommends repeat urinalysis after the treatment of other causes, and subsequent renal function testing, cystoscopy and imaging if symptoms do not resolve after treatment.[46] However, the low diagnostic yield of NVH evaluation and the frequency of other benign causes (such as infection) in everyday clinical practice may affect guideline adherence, and contribute to diagnostic delay. Future research should examine risk stratification based not only on presence of symptoms, but also on their severity, chronicity, recurrence or persistent nature. For example, guidelines should address cut-offs for degree of NVH that should warrant active reviews after treatments for UTIs; or cut-offs for number of UTIs treated before referral), while also taking into account cost-effectiveness

 Zhou Y, et al. BMJ Open 2019;9:e029143. doi:10.1136/bmjopen-2019-029143

of any follow-up actions. Emerging urine biomarkers and risk prediction tools may also be useful additions to improve diagnostic yield.[47 48]

## Concomitant benign conditions

The challenge in clinical practice arises when a presenting symptom may be the result of concomitant benign disease or cancer. In these cases, it is likely that a significant proportion of patients will be appropriately treated and reviewed, averting unnecessary investigations.

The observation that a UTI diagnosis delays cancer diagnosis in patients first seen by both urologists and non-urologists[37] suggests that diagnostic reasoning is challenging for clinicians in such situations. The PPV of a symptom for cancer probably falls if the patient has concomitant diseases which cause the symptom. Some evidence suggests that conditions such as benign prostate disease and kidney stones also delay the diagnosis of kidney cancer.[37] Whether and how much of this diagnostic delay is avoidable is yet to be determined, and should be a priority for future research. While current UK guidelines recommend that patients with persistent or recurrent UTIs should be referred for further evaluation, there is no US equivalent, nor guidelines on the management of other concomitant benign conditions and possible cancer. When no guideline exists, or adherence is not possible, close follow-up to ensure improvement or resolution of symptoms should take place, and patients should be instructed to return if symptoms do not improve.

## Clinician and system factors

Patients experienced shorter diagnostic intervals if they first presented to a urologist instead of another specialty doctor. This is not surprising given that urologists are likely to have better access to investigations, typically consisting of cystoscopy and upper renal tract imaging. The variations seen in evaluation and referral between other clinical specialties may indicate different levels of guideline awareness and adherence, although this evidence is scarce and inconsistent.

Process or system delays, such as patient non-attendance at appointments, delays in scheduling of appointments or non-receipt of referrals, all contribute to diagnostic delay,[17 24 33 49] although the magnitude of their effects is unclear. In addition to improving process and workflow issues within primary and secondary care services, wider system changes such as providing direct access to imaging and streamlining referral processes may also play a role in expediting cancer diagnosis. Diagnostic pathways such as the fast-track referral system for patients with alarm symptoms in the UK, or a telephone hotline service, may shorten primary care and total healthcare interval, although the cost-effectiveness of such pathways need to be evaluated in specific health system contexts.

## Strengths and limitations

Our review is the first to examine the evidence relating to factors affecting the quality of the diagnostic process in patients with bladder and kidney cancer. It builds on a previous review examining haematuria assessment in bladder cancer patients,[9] and looks at a range of urological symptoms, and also patients with kidney cancer. Although some of the studies did not adjust for all the confounders, the descriptive sections related mainly to the diagnostic intervals and appropriate statistical analyses were performed for examining the factors affecting these intervals, where relevant.

Unfortunately, we were unable to perform a meta-analysis due to the heterogeneity in study designs and outcomes. We were also unable to check the veracity of the comorbid disease labels in the papers that used coded information. All the studies are from high-income countries, and therefore may be less generalisable to other countries with differing healthcare systems.

## CONCLUSION

We found lack of consistency in defining diagnostic quality, including timeliness, of bladder and kidney cancer, and insufficient exploration of population-based evidence related to clinician and system factors affecting the quality of the diagnostic process. Our review highlights the need to improve evaluation of haematuria, and to develop high-quality evidence to inform guidelines on specific thresholds for active follow-up of high-risk symptomatic patients, which could be incorporated into risk prediction tools and clinical decision support. Future research should also identify and target preventable delays, especially in the context of concomitant benign conditions. Identifying patients with evaluation delays through electronic algorithms may also improve outcomes and reduce the sex inequality in survival for these cancers. In sum, our review identifies several potential areas of missed opportunities in bladder and kidney cancer diagnosis that may be avoidable and amenable to targeted interventions.

**Author affiliations**
[1]Primary Care Unit, Department of Public Health and Primary Care, University of Cambridge, Cambridge, UK
[2]Houston Veterans Affairs Center for Innovations in Quality, Effectiveness and Safety, Michael E. DeBakey Veterans Affairs Medical Center, Houston, Texas, USA
[3]Department of Medicine, Baylor College of Medicine, Houston, Texas, USA
[4]University of Exeter Medical School, Exeter, UK
[5]Department of Epidemiology and Public Health, Health Behaviour Research Centre, University College London, London, UK

**Acknowledgements** The authors would like to thank Isla Kuhn, medical librarian at the University of Cambridge School of Clinical Medicine Medical Library, for her advice and assistance with the development of the search strategy.

**Contributors** YZ, GL and FMW designed the study. YZ developed and performed the search. YZ and MvM performed the data extraction with MvM. YZ drafted the manuscript. MvM, HS, WH, GL and FMW critically revised the article.

**Funding** YZ is supported by a Wellcome Trust Primary Care Clinician PhD Fellowship (203921/Z/16/Z). The authors WH and FMW are coprincipal investigators

and the authors. GL and HS are coinvestigators of the multi-institutional CanTest Research Collaborative funded by a Cancer Research UK Population Research Catalyst award (C8640/A23385). HS is additionally supported by the Houston VA HSR&D Center for Innovations in Quality, Effectiveness and Safety (CIN 13-413).

**Competing interests** None declared.

**Patient consent for publication** Not required.

**Provenance and peer review** Not commissioned; externally peer reviewed.

**Data availability statement** There are no data in this work.

**Open access** This is an open access article distributed in accordance with the Creative Commons Attribution 4.0 Unported (CC BY 4.0) license, which permits others to copy, redistribute, remix, transform and build upon this work for any purpose, provided the original work is properly cited, a link to the licence is given, and indication of whether changes were made. See: https://creativecommons.org/licenses/by/4.0/.

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
