## [Reviewer comments · BMJ Open]

ARTICLE DETAILS

TITLE (PROVISIONAL)	Quality of the Diagnostic Process in Patients Presenting with Symptoms Suggestive of Bladder or Kidney Cancer: A Systematic Review
AUTHORS	Zhou, Yin; van Melle, Marije; Singh, Hardeep; Hamilton, Willie; Lyratzopoulos, Georgios; Walter, Fiona

VERSION 1 – REVIEW

REVIEWER	Dr Khaled Ahmed University of Birmingham, Birmingham Clinical Trials Unit (BCTU) Public Health building, Birmingham United Kingdom.
REVIEW RETURNED	28-Feb-2019

GENERAL COMMENTS	I would like to thank the authors for writing a well thought out review which is very much needed and relevant. I have a few minor query: Although the author's do justify why the search started from 1st January 2000 due to their prior knowledge it would be good if they could justify this on a non subjective basis i.e. by inserting a reference which indicates for instance technological advances where not available or papers were not available etc.
--

REVIEWER	Mia Schmidt-Hansen NGA, RCOG, UK.
REVIEW RETURNED	25-Mar-2019

GENERAL COMMENTS	This is a well-written systematic review that makes an important contribution to the knowledge base for cancer diagnosis. I have only the following few comments of which comments 1 and 5 need to be addressed as a minimum: Method/search strategy and study inclusion: 1) Searching only Medline and Embase is very restricted. As a minimum this needs to be complemented by a search of the Cochrane library too. 2) Rationale for restricting to 2000 onwards not very strong and suggests you have missed some pre-2000 studies (even if only few). Could it be improved, e.g., has the technology used in the diagnostic process changed significantly from 2000? Method/data extraction and quality assessment: 3) From the abstract it is clear that quality appraisal is also performed by 2 authors independently, but this is not clear from the methods section. I suggest including this information here too as it is a strength.
--

	Results: 4) The positive association between haematuria visits and diagnostic delay, I imagine reflects that the more haematuria visits a patient has, the longer the delay is, but I would suggest writing this out in full for clarity, especially in the context of the dose-dependent relationship between haematuria visits and likelihood of referral presented immediately beforehand. 5) You have clearly and explicitly assessed study quality and presented it in Table 1. Nevertheless you have not linked this to the results anywhere else in the results section. Some discussion of it in relation to the validity of the results is necessary in the results section, and I would also suggest adding a summary of that discussion to the actual Discussion section. In systematic reviews this needs to be explicit in all parts of the manuscript (apart from the introduction). 6) Instead of using NS to mean either non-specified (Appendix 2) or non-significant (Table 3), pick a different abbreviation for Appendix 2 such as NSp to avoid misunderstanding
--	--

REVIEWER	Shomik Sengupta Monash University Australia
REVIEW RETURNED	09-Apr-2019

GENERAL COMMENTS	This is a nice review of published literature on the timeliness and quality of investigation of symptoms of bladder and kidney cancer. The review is in narrative form, which makes synthesis of data somewhat more challenging than if a meta-analysis had been carried out - however, this seems to be a reflection of the heterogeneity of the literature rather than a limitation of the review itself.
---

REVIEWER	Jonathan Gelfond UT Health San Antonio, USA
REVIEW RETURNED	06-Jun-2019

GENERAL COMMENTS	The authors present an interesting systematic review of the literatures on the diagnostic evaluation for bladder and kidney cancer. The paper is clearly written. The authors were focused on the quality and the timeliness of the diagnostic evaluation. The applied NICE criterion was appropriate. The authors present a summary of previous findings on the factors associated with lower quality assessment and delay in diagnosis (female gender, urologist vs. general practitioner, etc.). The main findings are nicely summarized in Table 3. There are a couple of minor concerns.  1. Would it have been possible to do a meta-analysis of a subset of the studies or the outcome that were most homogenous? 2. There was over 15 years of data, which is a strength of the paper, but are there any time trends in these findings? 3. The studies discussed were mostly retrospective, but there were some prospective intervention studies that were targeting improvements in such diagnostic evaluations. In Table 2 the control vs intervention times in the right columns seem reversed. Could this be made more clear? 4. Could Table 2 identify the significant differences between men and women?
---

VERSION 1 – AUTHOR RESPONSE

Reviewer: 1 Please leave your comments for the authors below I would like to thank the authors for writing a well thought out review which is very much needed and relevant. I have a few minor query: Although the author's do justify why the search started from 1st January 2000 due to their prior knowledge it would be good if they could justify this on a non subjective basis i.e. by inserting a reference which indicates for instance technological advances where not available or papers were not available etc.	Thank you Reviewer 1 for your positive comment. We have cited the 2017 Ngo review on diagnostic timeliness of haematuria, which found 2 (relatively small) studies in the 1980s from Sweden and Denmark. However, we have supplemented our rationale for focusing on the year 2000 and beyond, mainly due to the knowledge that the two-week-wait/fast-track referral system was introduced in 2000, and that clinical practice, diagnostic quality and timeliness would have been affected by this. “We restricted our search to studies published from 2000 due to prior knowledge that there were few early relevant studies⁹, and that the quality of the diagnostic process for cancer might have been affected by the introduction of national initiatives such as the fast-track referral pathways in the United Kingdom (UK) in 2000.”
Reviewer: 2 This is a well-written systematic review that makes an important contribution to the knowledge base for cancer diagnosis. I have only the following few comments of which comments 1 and 5 need to be addressed as a minimum: Method/search strategy and study inclusion: 1) Searching only Medline and Embase is very restricted. As a minimum this needs to be complemented by a search of the Cochrane library too. 2) Rationale for restricting to 2000 onwards not very strong and suggests you have missed some pre-2000 studies (even if only few). Could it be improved, e.g., has the technology used in the diagnostic process changed significantly from 2000? Method/data extraction and quality assessment: 3) From the abstract it is clear that quality appraisal is also performed by 2 authors independently, but this is not clear from the methods section. I suggest including this information here too as it is a strength. Results: 4) The positive association between	Thank you Reviewer 2 for your helpful and constructive feedback. We have made several substantial improvements to the study, especially with regards to the search strategy, as detailed below. 1) We have now performed a search of the Cochrane library from inception to 13th June 2019, in addition to updating the original search to 13th June 2019. 2) As per response to Reviewer 1, we have strengthened the rationale for performing our search from 2000. 3) We have added this in Methods, Data Extraction and Quality assessment: “Quality appraisal was performed using a modified version of the Critical Appraisal Skills Programme (CASP) checklist for cohort studies by both authors (Table 1 Footnote¹³.” 4) We have amended the sentence in question under

haematuria visits and diagnostic delay, I imagine reflects that the more haematuria visits a patient has, the longer the delay is, but I would suggest writing this out in full for clarity, especially in the context of the dose-dependent relationship between haematuria visits and likelihood of referral presented immediately beforehand. 5) You have clearly and explicitly assessed study quality and presented it in Table 1. Nevertheless you have not linked this to the results anywhere else in the results section. Some discussion of it in relation to the validity of the results is necessary in the results section, and I would also suggest adding a summary of that discussion to the actual Discussion section. In systematic reviews this needs to be explicit in all parts of the manuscript (apart from the introduction). 6) Instead of using NS to mean either non-specified (Appendix 2) or non-significant (Table 3), pick a different abbreviation for Appendix 2 such as NSp to avoid misunderstanding	Results, Presenting Symptoms, 2nd paragraph last sentence: "...and a high-quality US study involving claims review found that increasing number of haematuria visits was associated with diagnostic delay for bladder cancer patients ²⁴." 5) We described the size and type of the studies in the Results section, but have now also added a paragraph about the quality of the studies in the Results section: The main bias and applicability concerns related to the suboptimal identification and/or adjustment for confounders in 18 of the studies, 6 of which were studies using questionnaires¹⁴⁻¹⁹, 10 were retrospective cohort studies providing descriptive statistics mainly, using record reviews (n=5)²⁰⁻²⁴ and electronic health records (n=5)²⁵⁻²⁹, 1 was a case-control study³⁰ and 1 an ecological study³¹. We also added a sentence to first paragraph of the Strengths and Limitations section, within Discussion: "Although some of the studies did not adjust for all the confounders, the descriptive sections related mainly to the diagnostic intervals and appropriate statistical analyses were performed for examining the factors affecting these intervals, where relevant." 6) We have changed NS in Appendix 2 to "Either" to avoid any confusion.
Reviewer: 3 This is a nice review of published literature on the timeliness and quality of investigation of symptoms of bladder and kidney cancer. The review is in narrative form, which makes synthesis of data somewhat more challenging than if a meta-analysis had been carried out - however, this seems to be a reflection of the heterogeneity of the literature rather than a limitation of the review itself.	Thank you for your feedback.

Reviewer: 4 The authors present an interesting systematic review of the literatures on the diagnostic evaluation for bladder and kidney cancer. The paper is clearly written. The authors were focused on the quality and the timeliness of the diagnostic evaluation. The applied NICE criterion was appropriate. The authors present a summary of previous findings on the factors associated with lower quality assessment and delay in diagnosis (female gender, urologist vs. general practitioner, etc.). The main findings are nicely summarized in Table 3. There are a couple of minor concerns. 1. Would it have been possible to do a meta-analysis of a subset of the studies or the outcome that were most homogenous? 2. There was over 15 years of data, which is a strength of the paper, but are there any time trends in these findings? 3. The studies discussed were mostly retrospective, but there were some prospective intervention studies that were targeting improvements in such diagnostic evaluations. In Table 2 the control vs intervention times in the right columns seem reversed. Could this be made more clear? 4. Could Table 2 identify the significant differences between men and women?	Thank you for your positive feedback and summary of our main findings. 1. We did have a look at performing a meta-analysis of the effect of sex and age on diagnostic quality/timeliness as these were the factors that were explored by the most number of studies and produced the most consistent results (in the case of sex at least). Unfortunately, there was too much heterogeneity in the outcomes and methodologies used. 2. Thank you for making an interesting point. Exploring the time trend would have been challenging to do in this study, as the included studies although included data spanning more than 20 years (including 1992), individual studies often reported findings over a long period spanning 5-10 years at times, without results for individual years. Exploring the time trend of clinical practice would certainly have been interesting, but was not feasible in this review. 3. Thank you for pointing our mistake in the original table 2. We have now changed the control and intervention times to reflect the accurate interval days for each group. 4. We have now indicated the significant differences in the diagnostic intervals between men and women, using an asterisk behind those with a p-value ≤ 0.05 to denote such situations.

VERSION 2 – REVIEW

REVIEWER	Mia Schmidt-Hansen National Guideline Alliance, Royal College of Obstetricians and Gynaecologists, UK
REVIEW RETURNED	03-Jul-2019
GENERAL COMMENTS	The revisions have addressed my comments. I recommend the manuscript is accepted.